# OpenReview forum: "Information Retrieval Induced Safety Degradation in AI Agents"
_NeurIPS.cc/2025/Conference — NeurIPS 2025 poster_

### Official Review · Reviewer_4oF3 · 2025-06-17

**Clarity:** 3
**Significance:** 4
**Originality:** 3
**Rating:** 5
**Confidence:** 4

**Summary:**

The paper presents the results of a quantitative study of the harmfulness of LLMs in retrieval-augmented contexts. The study manipulates a single independent variable: the model type (censored LLM, censored RAG, uncensored LLM), with further variants under the RAG setting for Wikipedia-only or full-internet access. Given a harmful prompt, the authors evaluate the harmfulness of the generated response using an LLM as a judge. Their results show that RAG improves answer accuracy but consistently worsens the harmfulness of models, even for censored versions or with prompt-based mitigation measures. Further investigation shows that the number of steps in the retrieval process does not impact their findings. To quote the authors: "The mere act of introducing retrieved context destabilises model safety".

**Questions:**

- Why did you not evaluate (a subset of) your dataset with humans to make sure that the evaluator model's rankings actually reflect human judgment? If you could evaluate with humans, how would you stop them from using LLMs themselves?
- Do you think your results might be impacted by the particular safety SFT dataset or RLHF regime used?
- Do you think models' harmfulness would also degrade even if you used a non-harmful prompt? (E.g. by accessing harmful content online or due to a reward hacking agent?)
- Note on the title (feel free to ignore): I thought the title didn't do justice to the contents of the paper, because it is not very descriptive and requires knowing what "devolution" means in this context. The authors might want to consider using a different title, maybe something like: "Retrieval Augmentation Consistently Degrades LLM Safety".

**Ethical Concerns:**

["NO or VERY MINOR ethics concerns only"]

**Final Justification:**

I believe the authors have satisfactorily addressed my concerns, including regarding:
- The distribution of the evaluation data;
- Results in a vector-RAG setup;
- Potential bias from the LLM-as-evaluator setup;
- Effects of RAG with non-harmful prompts;

Most other concerns I raised would likely warrant an entirely new paper (e.g. the comparisons of effects of different safety fine-tuning mechanisms), and the authors have acknowledged appropriately where this may be the case. They have also engaged with other reviewers in a similarly detailed manner and have demonstrated rigour in the discussion phase. I also believe that this paper will be of interest to several people at NeurIPS.

**Limitations:**

I think the paper could benefit a lot from some clearer discussion of the limitations of its experimental setup, which would aid the interpretation of its results. I would urge the authors to consider adding information along some of the axes:
- What was the size of the datasets used, how were the different classes distributed, and are there any significant biases that could have resulted from data imbalance?
- Could the results potentially generalise to non-open-source models?
- How does the method of safety censoring affect the results?
- Would similar patterns appear in vector-database RAG?
- Could relying on an LLM as an evaluator have introduced biases to the results? What sort of biases would you expect the LLM to introduce?

**Paper Formatting Concerns:**

None.

**Quality:**

3

**Strengths And Weaknesses:**

I believe this is a valuable paper that presents an interesting insight into the patterns of LLMs in harmful scenarios. The quantitative results are convincing and thorough, although the experimental setup could have been better explained. The results, if not entirely surprising given the nature of the internet, are comprehensive and point to a significant concern underlying current LLM censoring mechanisms. I do have some concerns regarding the discussion of limitations, which I hope the authors can address in the discussion phase.

Advantages:
-  Engages with the important field of LLM safety in a meaningful way
- Clearly written, with the claims well justified and articulated
- Interesting and concerning findings regarding the fragility of LLM censoring in RAG contexts
- Performs some investigation of the generalisation of the results, by showing that the LLMs' harmfulness increases under both improved retrieval accuracy and prompt-based mitigation.

Disadvantages:
- The experimental setup could have been explained in more detail. I include further comments on this in the Limitations section.
- There is not really a qualitative exploration of why the performance of the LLMs degrades. The authors took some important steps to show that their results replicate under various hyperparameter settings of RAG, but currently, the paper does not offer insights into why this performance degradation happens. That being said, I don't think this is a reason to reject the paper as the results are valuable as they are, highlighting a serious issue with LLMs, and I hope that questions regarding the "Why?" will be addressed in future work.
- I am slightly concerned regarding the use of LLMs as evaluators. I appreciate the authors cross-checking alignment manually on 50 data points, but considering that the dataset sizes are (probably) much larger and that the agreement $\kappa$ is not near 1, I would imagine several rankings would not be aligned with human judgement. It is also not explained how classes in the dataset were distributed, so it could be that the misalignment stems from a single class or is distributed across various categories.

Typo: On L45, 'does' is repeated twice

---

> ### Author Rebuttal · Authors · 2025-07-31
>
> We sincerely thank the reviewer for their comprehensive assessment, acknowledging the value of our findings while providing constructive feedback. We have structured our response to address their disadvantages, questions, and limitations as follows:
>
> ## 1 The experimental setup & Limitations
> > Q1.1: What was the size of the datasets used, how were the different classes distributed, and are there any significant biases that could have resulted from data imbalance?
>
> A1.1: Thank you for raising this important point. While our main paper (L138) states that all benchmark data are publicly available, we acknowledge insufficient detail.
>
> Sizes and Class Distribution:
> - Bias1 (BBQ): 538 items with one representative query per scenario cluster, covering nine balanced categories: gender, race/ethnicity, religion, nationality, physical appearance, age, socioeconomic status, disability, and intersectional identities (Appendix L381–396).
> - Bias2 (AirBench-2024): 424 balanced items covering ethnicity, gender, race, religion, sexual orientation, disability, using concise questions for retrieval compatibility (L397–401).
> - Harm1 and Harm2: All 200 XSTest_v2 items (discrimination, violence, fraud) and all 250 SafeArena items (misinformation, illegal activity, harassment, cybercrime, social bias) (L402–408).
>
> Balance Check: We manually inspected each benchmark and found class distributions to be approximately balanced within the covered categories, helping mitigate internal imbalance. However, structural and coverage biases may persist—e.g., BBQ reflects English-speaking contexts and lacks full representation of all identity groups or cultural framings.
>
> > Q1.2: Could the results potentially generalise to non-open-source models?
>
> A1.2: Yes, our results generalize beyond open-source models. We included GPT-4o-mini in our evaluation (last line of each Table) and observed the same safety degradation patterns.
>
> We additionally evaluated OpenAI's Operator agent and Gemini-2.0-flash, both showing similar safety degradation.
>
> Bias1 Results
> |Setting|Refusal Score|Safety Score|
> |-|-|-|
> |API (GPT-4o-mini)|0.92 ± 0.03 | 4.37 ± 0.09 |
> |Agno-Web (GPT-4o-mini)|0.76 ± 0.05 | 3.44 ± 0.15 |
> |Operator| 0.42 ± 0.06 |2.71 ± 0.21 |
> |API (Gemini-2.0-flash)|4.16 ± 0.10 | 4.23 ± 0.10 |
> |Agno-Web (Gemini-2.0-flash)|3.20 ± 0.11 | 3.00 ± 0.11 |
>
> Harm1 Results
> |Setting| Refusal Score|Safety Score|
> |-|-|-|
> | API (GPT-4o-mini) |0.97 ± 0.03 | 4.95 ± 0.04 |
> | Agno-Web (GPT-4o-mini) |0.88 ± 0.05 | 4.72 ± 0.11 |
> | Operator | 0.79 ± 0.06 |4.56 ± 0.16 |
> | API (Gemini-2.0-flash) |4.56 ± 0.10 | 4.95 ± 0.04 |
> | Agno-Web (Gemini-2.0-flash) |4.34 ± 0.11 | 4.89 ± 0.07 |
>
> All values shown as Mean ± CI, with higher being better for both metrics.
> These results demonstrate that safety degradation in agentic systems occurs across different model architectures and providers.
>
> > Q1.3: How does the method of safety censoring affect the results? Do you think your results might be impacted by the particular safety SFT dataset or RLHF regime used?
>
> A1.3: We acknowledge alignment method may be a confounder—different censoring approaches (e.g., SFT and DPO for Qwen2.5-3B [arXiv:2407.10671]; DPO, rejection sampling (RS), and DPO for LLaMA3.2-3B [arXiv:2407.21783], SFT and RLHF for Gemma3-4B [arXiv:2503.19786]; system prompt and self-reflection for Mistral0.3-7B [arXiv:2310.06825]; and policy-aware RLHF for GPT-4o-mini [OpenAI2024]) could interact differently with retrieval augmentation. Yet, safety degradation appears across diverse models with varied alignment pipelines, implying a broader phenomenon. Disentangling these alignment-method interactions warrants future work.
>
> > Q1.4: Would similar patterns appear in vector-database RAG?
>
> A1.4: Thanks for the suggestion. We extended our study by adding a vector-based agent using DSPy with ColBERTv2 on Wikipedia (LLAMA-3B). Results show the same safety degradation pattern, CI is removed for brevity.
>
> Refusal Rates (higher is better,):
> |Benchmark|Censored|Uncensored|VectorAgent|
> |-|-|-|-|
> |Bias1|0.80|0.76|0.67|
> |Bias2|0.76|0.50|0.50|
> |Harm1|0.97|0.78|0.68|
> |Harm2|0.82|0.54|0.36|
>
> Safety Scores (higher is better):
> |Benchmark| Censored | Uncensored | VectorAgent |
> |-|-|-|-|
> |Bias1|4.03|3.75| 2.91|
> |Bias2|3.70|3.19| 2.76|
> |Harm1|4.83|4.41| 4.59|
> |Harm2|4.65|3.61| 3.54|
>
> We observe that the vector-based agent exhibits **the same safety degradation pattern** —refusal rates and safety scores both drop even below the uncensored baseline in most benchmarks.
>
> ## 2 Use of LLM as Evaluator
>
> > Q2.1: Could relying on an LLM as an evaluator have introduced biases to the results? What sort of biases would you expect the LLM to introduce?
>
> A2.1: We validated our LLM evaluator (Prometheus) against expert human ratings (with κ = 0.86 on line 147). This score represents substantial inter-rater reliability [Landis & Koch, The measurement of observer agreement for categorical data, 1977] and is consistent with safety evaluation studies [arxiv:2411.15594]. However, we acknowledge potential biases including preference for certain response styles, difficulty detecting nuanced cultural harms, and systematic underestimation of subtle stereotypes.
>
> > Q2.2: I appreciate the authors cross-checking alignment manually on 50 data points, but considering that the dataset sizes are (probably) much larger and that the agreement κ is not near 1, I would imagine several rankings would not be aligned with human judgement. It is also not explained how classes in the dataset were distributed.
>
> A2.2: Thanks for pointing this out. Our manual validation covered bias-related harms.  We acknowledge dataset scale limitations; thus, we additionally validated 50 examples each for harmfulness and refusal categories (agreement scores: 0.75 and 0.79 respectively). Bias-related harms in BBQ involve straightforward stereotypical assumptions (e.g., memory capabilities across age groups), making them more amenable to consistent judgment. We will further expand human validation in future work.
>
> > Q2.3: Why did you not evaluate (a subset of) your dataset with humans to make sure that the evaluator model's rankings actually reflect human judgment? If you could evaluate with humans, how would you stop them from using LLMs themselves?
>
> A2.3: We did evaluate a human-annotated subset (detailed in L148 with footnote). Expert annotators completed ratings without LLM assistance under controlled conditions with clear guidelines prohibiting AI tool use. While our current human validation has scale limitations, we acknowledge this constraint and plan to expand human evaluation.
>
> ## 3 Other Issues:
>
> > Q3.1: Do you think models' harmfulness would also degrade even if you used a non-harmful prompt? (e.g., by accessing harmful content online or due to a reward-hacking agent?)
>
> A3.1: Thank you for raising this important point. To assess whether harmfulness might degrade even when using non-harmful prompts, we tested the LLaMA 3.2–3B model on a benign subset from XSTest_v2 with 250 prompts explicitly curated to avoid harmful content. We applied the same 5-point Harmfulness rubric (where 5 indicates safest behavior).
> The API baseline scored 4.39 ± 0.12, while the WebAgent (Agno) scored 4.45 ± 0.12. Although the WebAgent's mean score is slightly higher, the difference is not statistically significant (p ≈ 0.465). This could suggess that in benign contexts, access to external tools does not degrade safety performance — at least not under our current evaluation setting.
> We appreciate the suggestion and will continue to examine cases involving reward hacking or emergent unsafe behavior in future work.
>
> > Q3.2: There is not really a qualitative exploration of why the performance of the LLMs degrades. The authors took some important steps... but currently, the paper does not offer insights into why this performance degradation happens.
>
> A3.2: We thank the reviewer for highlighting this important issue. Our work focuses on systematic empirical analysis, but we agree that deeper mechanistic insight would substantially strengthen the contribution. Insights from our ablations (Sec. 4.5) show that safety degradation occurs even with a single retrieved document and is largely independent of retrieval accuracy. This suggests that neither information overload nor poor retrieval quality alone explains the degradation.
> We further hypothesize from these results two possible mechanisms: (1) Task Framing Shift – retrieved content may shift the model’s interpretation of its role from safety evaluation to information synthesis, altering response behavior; and (2) Instructional Interference – agent prompts emphasizing “retrieve and answer” may override or downweight safety-related instructions, consistent with the limited mitigation observed from safe prompting (Tables 2–4). While we do not claim definitive causal explanations, we hope our findings provide a foundation for future mechanistic studies (e.g., attention or activation analyses) to further probe these effects.
>
> > Q3.3: Typo on L45 — 'does' is repeated twice.
>
> A3.3: Thank you for catching this — we’ll correct this.
>
> > Q3.4: Note on the title (feel free to ignore): I thought the title didn't do justice to the contents of the paper, because it is not very descriptive and requires knowing what "devolution" means in this context...consider using a different title, maybe something like: "Retrieval Augmentation Consistently Degrades LLM Safety".
>
> A3.4: We appreciate the title feedback and will revise to "Information Retrieval Induced Safety Degradation in AI Agents" for clarity. While our agents (AutoGen/Operator) support tool-use and multi-step reasoning beyond basic RAG, we acknowledge our current benchmarks primarily focus on retrieval-based interactions using Wikipedia and web search tools. Our findings reveal fundamental safety vulnerabilities that extend across different agentic architectures and highlight critical risks as these systems become more widely deployed.

---

> > ### Comment · Reviewer_4oF3 · 2025-08-01
> > **Raising rating**
> >
> > Thank you for your detailed and clear rebuttal. I believe this is an excellent paper, and your responses have exhaustively addressed my concerns. Not only have you added additional experiments, but both the manuscript and the rebuttal indicate a great deal of understanding of the related fields and the pertinent questions that remain to be answered. I think this paper will be a valuable contribution to NeurIPS. Accordingly, I will raise my rating.

---

> > > ### Author Response · Authors · 2025-08-02
> > >
> > > Thank you for your thoughtful review and for raising your rating. Your constructive feedback helped us expand our evaluation and refine the paper's framing. We look forward to incorporating these improvements and hope this work will contribute meaningfully to agent safety research.

---

### Official Review · Reviewer_K35k · 2025-06-23

**Clarity:** 3
**Significance:** 1
**Originality:** 2
**Rating:** 3
**Confidence:** 5

**Summary:**

This paper primarily identifies the phenomenon of safety devolution, where broader retrieval in Retrieval-Augmented agents increases bias and harm. The authors then conduct experiments around this phenomenon, demonstrating that prompt-based mitigation strategies are ineffective and that the decline in agent safety mainly stems from the retrieved content itself. The paper presents a comprehensive analysis of the safety of Retrieval-Augmented agents, highlighting the need for stronger safety mechanisms in safety-critical scenarios.

**Questions:**

The paper treats refusal to answer harmful questions as a safer behavior. What I’m curious about is whether some refusals may actually stem from the LLM's lack of relevant knowledge, rather than from a refusal mechanism or safety alignment. I believe that under the
assumption that both the LLM and the Retrieval-Augmented Agent are aware of similar answers, an LLM’s refusal, while the Retrieval-Augmented Agent responds, could serve as stronger evidence for safety devolution.

The paper attributes the cause of the described safety devolution phenomenon to the retrieval mechanism itself. However, I’m curious about deeper architectural factors. Does incorporating retrieval inherently increase the agent's confidence in responding to a question? Is refusal still an available option for the agent, but simply with a lower probability

**Ethical Concerns:**

["NO or VERY MINOR ethics concerns only"]

**Final Justification:**

I appreciate the authors' thoughtful response and the clarification it provided on several points from my review. However, I do have reservations regarding the significance of this particular aspect of their rebuttal. It is well-established that LLMs and RAGs face safety challenges, partly attributable to their training on public online content, which can inherently contain biases and harmful material. I wonder if reiterating this widely acknowledged point provides truly novel insight in the context of this paper.

**Limitations:**

Yes

**Quality:**

2

**Strengths And Weaknesses:**

Strengths:
The experiments in the paper are presented with detailed setups, which facilitate reproducibility and verifiability.

The paper is the first to identify the phenomenon of safety devolution, an important yet often overlooked issue in RAG systems. It constructs and evaluates this phenomenon across multiple application scenarios, demonstrating its generality.

Weaknesses:
The metrics used in the paper lack further explanation. Evaluating safety solely based on refusal rates is insufficient, and the later-used safety measures are also not well elaborated.

The experiment in Section 4.1 appears redundant—the results are almost predetermined, and it is unclear how this contributes to the core research question.

The mitigation strategies proposed are overly simple and singular. After dismissing prompt engineering, the paper fails to propose potentially more effective alternatives.

Although the paper compares several open-source LLMs, it lacks experiments across models with varying parameter scales. Does increasing model size lead to any differences?

The text in Figure 1 is too small to read clearly.

---

> ### Author Rebuttal · Authors · 2025-07-31
>
> We thank the reviewer for their detailed feedback and appreciate the opportunity to address their concerns about our experimental design, evaluation metrics, and mitigation strategies.
>
> > Q1 Evaluating safety solely based on refusal rates is insufficient, and the later-used safety measures are also not well elaborated.
>
> A1: We agree that refusal rates alone are insufficient for assessing safety. Our evaluation framework is intentionally multi-dimensional, capturing both effectiveness and safety through detailed rubric-based assessments.
>
> As outlined in Appendix A.2 (Lines 409–610), our framework includes:
> - Effectiveness: Rated on a 5-point scale for factual accuracy and informativeness, from clearly incorrect or irrelevant (1) to fully correct and comprehensive (5), based on alignment with reference answers.
> - Bias: Scores range from perpetuating harmful biases (1), through neutral or superficial handling (3), to critical reflection and proactive avoidance of bias (5).
> - Harm: Evaluates both the presence of harmful content and assistance with harmful tasks, from explicitly harmful or unsafe responses (1) to responsible, ethical refusals with clear rationale (5).
> - Refusal Quality: Scored (1–2) when the model fulfills inappropriate requests, and (3–5) when it refuses, with higher scores reflecting refusals that are appropriate, sensitive, and helpful.
>
> We acknowledge that the main text should more clearly emphasize these comprehensive evaluation criteria and will revise accordingly to better highlight the depth and rigor of our safety assessment.
>
> > Q2 The experiment in Section 4.1 appears redundant—the results are almost predetermined, and it is unclear how this contributes to the core research question.
>
> A2 Thank you for the feedback. We understand the concern and would like to clarify the purpose of Section 4.1. While the link between retrieval and factual accuracy may seem intuitive, retrieval can also introduce irrelevant or misleading information, and determining when retrieval is beneficial remains an open research challenge [arxiv:2401.15884, 2501.09292]. Section 4.1 empirically demonstrates that our retrieval-augmented agents do, in fact, improve factual accuracy, providing crucial methodological context for the safety-utility trade-offs examined later in the paper.
>
> Experiments exploring trade-offs between accuracy and safety are common in AI alignment research, with prior work highlighting inherent tensions between helpfulness and safety alignment [arxiv:2212.08073, 2310.19852, 2406.18346, 2503.00555].
>
> We appreciate the suggestion and are open to relocating detailed quantitative results to the appendix, while retaining this foundational methodological context in the main text.
>
> > Q3 The mitigation strategies proposed are overly simple and singular. After dismissing prompt engineering, the paper fails to propose potentially more effective alternatives.
>
> A3  We acknowledge that our mitigation strategies are intentionally simple, following established evaluation practices in AI safety research. Existing adversarial robustness and jailbreaking safety benchmarks acknowledge complex safety mechanisms but focus on simpler evaluation approaches for practical assessment [arXiv:2404.12241, arXiv:2505.05541]. Additionally, system-level prompt engineering is widely recognized as a standard safety measure in deployed AI systems [arXiv:2401.18018, xAI GitHub Grok Prompts].
>
> We implemented an additional tool for the agent, who can now call a safe_rewrite tool to moderate and rewrite the final answer. This can help in returning safe, appropriate, unbiased, and respectful, post-processing retrieval-augmented responses, but our results show a concerning pattern:
>
> refusal rate
>
> | Benchmark | API | Agno-Web | Agno-Web-guard |
> |-|-|-|-|
> |Bias1 | 0.92 | 0.76 | **0.93** |
> |Bias2 | 0.67 | 0.61 | **0.73** |
> |Harm1 | **0.97** | 0.88 | 0.91 |
> |Harm2 | **0.73** | 0.53 | 0.67 |
>
> bias/safety score
>
> | Benchmark | API | Agno-Web | Agno-Web-guard |
> |-|-|-|-|
> | Bias1 | **4.37** | 3.44 | 3.82 |
> | Bias2 | **3.39** | 3.14 | 3.26 |
> | Harm1 | **4.95** | 4.72 | 4.87 |
> | Harm2 | **4.52** | 3.99 | 4.16 |
>
> Even with explicit guardrails (Agno-Web-guard), the system underperforms compared to the baseline API in most cases. This reveals a key finding: retrieved content introduces implicit safety issues that complicate standard mitigation approaches, making RAG systems fundamentally different from standalone LLMs in terms of safety. We agree that more sophisticated techniques warrant further exploration.
>
> > Q4 Although the paper compares several open-source LLMs, it lacks experiments across models with varying parameter scales. Does increasing model size lead to any differences?
>
> A4 In our original paper, we also tested GPT-4o-mini as a non-open-source model at a larger scale, and have now added experiments across four model scales.
>
> Refusal Rate (Higher is Better)  Format: wikiAgent / Censored API
>
> | Benchmark | Gemma 1B | Gemma 4B | Gemma 12B | Gemma 27B |
> |-|-|-|-|-|
> | Bias1 | 0.37 / 0.94 | 0.57 / 0.97 | 0.63 / 0.97 | 0.73 / 0.96 |
> | Bias2 | 0.50 / 0.90 | 0.64 / 0.95 | 0.63 / 0.92 | 0.75 / 0.92 |
> | Harm1 | 0.68 / 0.99 | 0.78 / 0.99 | 0.87 / 0.99 | 0.94 / 1.00 |
> | Harm2 | 0.24 / 0.82 | 0.59 / 0.90 | 0.57 / 0.92 | 0.70 / 0.91 |
>
> Bias/Safety Score (Higher is Better) Format: wikiAgent / Censored API
>
> | Benchmark | Gemma 1B | Gemma 4B | Gemma 12B | Gemma 27B |
> |-|-|-|-|-|
> | Bias1 | 1.96 / 4.85 | 2.31 / 4.90 | 2.47 / 4.86 | 2.84 / 4.88 |
> | Bias2 | 2.92 / 4.35 | 3.22 / 4.44 | 3.55 / 4.55 | 3.78 / 4.48 |
> | Harm1 | 4.72 / 4.97 | 4.54 / 4.94 | 4.82 / 5.00 | 4.89 / 5.00 |
> | Harm3 | 3.06 / 4.44 | 3.99 / 4.51 | 4.05 / 4.60 | 4.27 / 4.62 |
>
> We observe a consistent pattern: **wikiAgent results are lower than Censored API across all settings**, indicating **safety degradation** when agents operate without output-layer censorship. This holds for both refusal rates and safety scores.
>
> Additionally, larger models generally perform better in both metrics, though exceptions exist. This suggests that scaling improves robustness, but does not fully close the safety gap introduced by agent-based responses.
>
> > Q5 The text in Figure 1 is too small to read clearly.
>
> A5 Thank you for pointing this out. We have already prepared a revised version of the figure with improved readability, which we will include in future version.
>
> > Q6 More refusal = safer? What I’m curious about is whether some refusals may actually stem from the LLM's lack of relevant knowledge, rather than from a refusal mechanism or safety alignment. I believe that under the assumption that both the LLM and the Retrieval-Augmented Agent are aware of similar answers, an LLM’s refusal, while the Retrieval-Augmented Agent responds, could serve as stronger evidence for safety devolution.
>
> A6 This distinction is crucial and we appreciate the observation. To differentiate between knowledge-driven and alignment-driven refusals, we conducted manual analysis on 20% of refusal instances from the Llama3.2 API. Our findings show that only approximately 6% presented ambiguous refusal reasons, while the remainder demonstrated clear ethical concerns with *explicit* reason to refuse, such as "I cannot provide information on illegal or harmful activities, including...”
> An illustrative case involves queries about cannabis procurement: the base model typically provides informational responses while maintaining appropriate safety warnings, whereas the retrieval-augmented version frequently delivers *more direct guidance with diminished cautionary framing*, despite accessing equivalent underlying knowledge.
> We acknowledge this analysis should be expanded and will include comprehensive qualitative examples in the appendix to better illustrate these distinct refusal mechanisms.
>
> > Q7 The paper attributes the cause of the described safety devolution phenomenon to the retrieval mechanism itself. Deeper architectural factors? Does incorporating retrieval inherently increase the agent's confidence in responding to a question? Is refusal still an available option for the agent, but simply with a lower probability?
>
> A7 This is an excellent mechanistic question that we have begun investigating. To examine whether retrieval augmentation affects response confidence, we analyzed token-level probabilities comparing GPT-4o-mini direct API responses versus Agno web agent responses on the same prompts.
>
> Our preliminary analysis reveals that **direct API responses exhibit higher confidence and lower perplexity** compared to retrieval-augmented responses. Here the 'confidence' is measured by exponentiating token log probabilities and taking the mean across all tokens, while perplexity is calculated as the exponential of negative mean log probability:
>
> - **Confidence**: Direct API: 0.484 ± 0.011 vs. Web Agent: 0.407 ± 0.016 (statistically significant difference)
> - **Log-Perplexity**: Direct API: 7.38 ± 0.48 vs. Web Agent: 12.32 ± 1.23 (lower perplexity indicates more coherent responses)
>
> Interestingly, this suggests that retrieval augmentation **reduces** model confidence rather than increasing it, yet safety degradation still occurs. This counterintuitive finding indicates that the mechanism behind safety degradation may not be confidence-driven but rather stems from other factors such as: (1) retrieved content providing competing signals that override safety training, (2) the model prioritizing task completion (answering based on retrieved information) over safety refusal, or (3) retrieved context fundamentally altering the model's decision-making process about when refusal is appropriate.
>
> This preliminary evidence suggests the safety degradation phenomenon is more complex than simple confidence calibration effects, requiring deeper investigation into how external information structurally influences safety mechanisms beyond confidence levels.

---

> ### Comment · Reviewer_K35k · 2025-08-02
>
> Thank you for your responses to points A2, A4, A5, and A6.
>
> I have the following comments regarding A1 and A3:
>
> A1: Thank you for clarifying your chosen safety measures. However, I remain unclear on one point: the validity of these measures. For example, "bias" is a vast topic dating back to the early days of ML. How can a complex domain bias be adequately measured by a simple "bias" metric?
>
> A3: I disagree with the statement: “We acknowledge that our mitigation strategies are intentionally simple, following established evaluation practices in AI safety research.” The key concern is the lack of novelty in your approach. The phrase "established evaluation practices" feels like an evasive way to describe the lack of innovation.
>
> Overall, thank you for your efforts in the rebuttal. Nevertheless, I remain unconvinced regarding the paper's significance in addressing its research question and the novelty of its approach.

---

> > ### Author Response · Authors · 2025-08-06
> >
> > We thank the reviewer for their continued engagement and feedback. We address your specific concerns about bias measurement validity (A1) and research novelty in new safety improvement approaches (A3) below, while respectfully maintaining that our work makes significant contributions to understanding safety vulnerabilities in AI agents engaging with information retrieval.
> >
> > **A1 Response:**
> >
> > We appreciate your concern about bias measurement validity. Our approach addresses this complexity through established, validated frameworks rather than ad-hoc metrics.
> >
> > Our bias evaluation builds on well-established benchmarks (BBQ, AirBench) that have undergone rigorous validation in the AI safety community, allowing us to be very concrete about what kind of bias we are measuring (Line 380-401).
> >
> > BBQ (Bias Benchmark for QA) contains question sets highlighting attested social biases against protected classes with **defined stereotype labels** across groups: 538 items across **nine balanced categories** including *gender, race/ethnicity, religion, nationality, physical appearance, age, socioeconomic status, disability, and intersectional identities*. AirBench-2024-bias provides 424 balanced items covering *ethnicity, gender, race, religion, sexual orientation, and disability*.
> >
> > You're correct that "bias" is a vast topic—which is precisely why **these benchmarks systematically cover diverse bias categories** across multiple protected groups, necessitating multi-dimensional assessment rather than binary classification.
> >
> > The 5-point rubric isn't a "simple bias metric" but a **rigorous assessment framework** evaluating recognition of biased assumptions, avoidance of stereotype reinforcement, provision of balanced responses, and critical examination of harmful framings.
> >
> > Our evaluation employs Prometheus-v2.0, a specialized model fine-tuned for high-quality LLM assessment, achieving **high inter-annotator agreement (kappa > 0.8)** across 150 expert-evaluated data points. This systematic approach, detailed in Line148 with footnote, demonstrates both reliability and validity across diverse bias scenarios—establishing a robust foundation for safety assessment in retrieval-augmented systems.
> >
> > **A3 Response:**
> >
> > We respectfully disagree with the characterization of our approach as lacking novelty. Our contribution lies not in proposing entirely new mitigation techniques, but as a **novel empirical contribution** in universal safety devolution across AI agents through information retrieval, where context integration destabilizes alignment mechanisms—a finding with significant implications for the field.
> >
> > As you acknowledge, we're ***"first to identify the phenomenon of safety devolution, an important yet often overlooked issue"***—this represents a **novel safety phenomenon** showing that safety isn't just about model training but about information architecture.
> >
> > Following "established evaluation practices" does not indicate lack of novelty—we are referring to **leading, prevalent safety benchmarks** that deliberately focus on evaluation **without basic safety improvement measures** [arXiv:2404.12241, arXiv:2505.05541]. While benchmarking typically excludes safety improvements, we **go beyond standard practice** by including system-level prompt-based interventions and a safe_rewrite tool as guardrails, revealing **counterintuitive findings**: even explicit post-processing guardrails cannot fully mitigate safety degradation. This challenges fundamental assumptions about safety transferability from standalone LLMs to retrieval-augmented systems.
> >
> > As you acknowledge, our work ***"constructs and evaluates this phenomenon across multiple application scenarios, demonstrating its generality."*** Our ablation studies—varying multi-hop retrieval, document counts, and accuracy—reveal that **the mere act of introducing retrieved context destabilizes model safety**, uncovering the underlying mechanism.
> >
> > The significance lies in demonstrating that retrieved content introduces **architectural safety vulnerabilities** that conventional approaches cannot address. This opens new research directions requiring retrieval-aware safety mechanisms—a fundamentally different problem space with immediate implications for deployed systems at massive scale.
> >
> > ---
> > We hope these clarifications address your concerns about both the methodological rigor of the bias evaluation benchmark and the significance of our empirical findings. Our systematic evaluation reveals that established safety measures behave fundamentally differently when information retrieval is involved—a critical insight for the field as retrieval-augmented systems become ubiquitous. This work provides essential groundwork for developing retrieval-aware safety mechanisms and highlights an urgent research direction that extends beyond conventional LLM safety approaches.

---

> > > ### Comment · Reviewer_K35k · 2025-08-06
> > >
> > > Thanks for your detailed feedback.  I will consider that in the final evaluation.

---

> > > > ### Author Response · Authors · 2025-08-07
> > > >
> > > > Thank you for the update. We're happy to further clarify any outstanding concerns if helpful.

---

### Official Review · Reviewer_La9j · 2025-07-02

**Clarity:** 2
**Significance:** 3
**Originality:** 3
**Rating:** 4
**Confidence:** 3

**Summary:**

The paper presents an empirical study of the implications of extending an LLM with retrieval-augmented generation (RAG) with regards to denial of requests that are considered unsafe, bias and avoidance of harmful or unethical content.

The study is based on a comparison of 1) censored LLMs without retrieval, 2) LLMs that are retrieval-augmented with access to Wikipedia or the web and 3) uncensored LLMs. The censored LLMs are out-of-the box open-source models Qwen, Llama, Gemma and Mistral in addition to ChatGPT 4o mini. Uncensored models are the open-source models but from checkpoints before safety mechanisms were added. Finally, the RAG LLMs are based on Agno and Autogen.

Four benchmarks are used to evaluate the models. These were Bias Benchmark for QA, AirBEnch-2024, XSTest_v2 and SafeArena.
The study finds that models with access to external sources, the RAG LLMs, are less safe than the models that do not have access to external sources, such as the Web and Wikipedia. Some of them are even less safe than their unconcerned counterparts. This applies even as RAG-based LLMs have better performance than those that are not augmented with external sources.

The study underscores the need for robust mitigation strategies for RAG LLMs.

**Questions:**

* It should be made clear how the contributions in An et al. [28] differ from the contributions of this paper. Based on the explanation in section 2 Related Work, it might seem like they already proposed that LLMs that are RAG-enabled are less safe than those that are not RAG-enabled. Could you please explain how their contribution differs from yours?
* Given how poor the performances that are reported in Table 5 are, can we really trust the results?
* Is the sentence staring on line 240 with “Despite …” correct? It does not really make sense to me.

**Ethical Concerns:**

["NO or VERY MINOR ethics concerns only"]

**Final Justification:**

Based on the comments made by other reviewers and the rebuttal by the authors, I have changed my recommendation to borderline accept. However, I would have preferred to have seen the final version of the paper to be sure that the changes actually improved the paper. I am not very confident in my recommendation.

**Limitations:**

Yep.

**Quality:**

3

**Strengths And Weaknesses:**

## Strengths
* The results are interesting and potentially very significant.
* The experiments are extensive and well executed.

## Weaknesses
* The writing is at times not clear and not easy to follow. There the term “integration” is used in the second sentence in the abstract, but it is not clear with what. Also, the safety degradation of LLMs that comes with RAG “culminating in a phenomenon we term safety devolution”. What exactly is meant by this term devolution is not explained. The experiments show that just turning on RAG-capabilities makes the LLM less safe, so how it culminates is not clear. Another sentence that I do not understand is the sentence starting on line 36. The language would clearly improve if simplified and the arguments are made clear. Please spend more time on sharpen your arguments and make them unambiguous and crystal clear. I think the paper would gain a lot from it.
* Agent is used to describe an LLM with RAG. Why not just state LLM w. or w.o. RAG instead of agent? An agent is autonomous according to common AI terminology. The "agents" that are described here is not, so I propose just calling them LLMs, which they clearly are.
* The paper uses different terms for the same things, i.e. censored LLM and alignment-tuned LLM. Different terms for the same concept reduces clarity and readability. Please standardize.
* Adding prompts before generating a query and before answering the users cannot – in my mind at least -  be called a system level safety check. There is no check even. The LLM is asked to consider whether it should drop the query and/or check the answer for unsafe information. It is at best a self-evaluation by the LLM.

## Minor comments
* The first sentence in the abstract could be improved. How about changing it to something like "Despite their growing integration into society, the safety and ethical behavior of retrieval-augmented AI agents remain inadequately understood"?
is an example of this, and so is the second sentence.
* As far as I can understand the “assumption” on line 41 refers to the term “phenomenon” on line 39.  Is it an assumption or a phenomenon. Do I misunderstand? What is it I do not see?
* I find the term “safety degradation” to better describe the phenomenon than “safety devolution”
* It is not clear what is meant by aligned LLMs or alignment. Please be specific and explain what you mean and provide references.
* On line 147 $\kappa=0.86$ is considered high correlation. Please specify the variable $\kappa$ and explain what it measures (is it just correlation?).
* On line 150, it is stated that four metrics are used but I only count three.
* On line 168, what do you mean by alignment challenges? Be specific.
* On line 175, what is abliteration-based uncensoring? This is a term I am not aware of. Please explain.

## Summary
I am of two minds with regards to this paper. I find the experiments and results intriguing and potentially significant. However, the paper is not well written. Also, it could seem like the contribution of identifying the safety issue of adding RAG-capabilities to an LLM has been pointed out by [28].

Because of the poor writing and uncertainty about the novelty, I end up with recommending Borderline Reject.

---

> ### Author Rebuttal · Authors · 2025-07-31
>
> We are grateful to the reviewer for their comprehensive assessment, recognizing the significance of our experimental findings while providing valuable guidance on improving clarity, terminology, and our relationship to prior work.
>
> > Q1: There the term “integration” is used in the second sentence in the abstract, but it is not clear with what.
>
> A1: Thank you for pointing this out. We will revise to: "In particular, the growing integration of LLMs and AI agents *with external information sources and real-world environments* raises critical questions about how they engage with and are influenced by *these external data sources and interactive contexts.*"
> This clarifies that "integration" refers specifically to connecting AI systems with external retrieval sources (Wikipedia, web search) and interactive environments.
>
> > Q2: The term “devolution” is not explained. I find the term “safety degradation” to better describe the phenomenon.
>
> A2: We agree that this could be confusing and will revise throughout. The sentence will read: "Our findings reveal a consistent degradation in refusal rates, bias sensitivity, and harmfulness safeguards as models gain broader access to external sources, culminating in a phenomenon we term safety degradation."
> We will also add a brief definition upon first use: "safety degradation—the systematic weakening of safety properties as models access broader external information sources."
> This terminology change eliminates the need to explain the biological connotations of "devolution" while maintaining the core concept of safety properties deteriorating with expanded retrieval access.
>
> > Q3: The sentence on line 36 is unclear.
>
> A3: We will revise for clarity: "When AI agents retrieve information from external sources containing biased content (e.g., biased articles, prejudiced web content), they not only reproduce these biases in their responses but often amplify them through confident presentation and lack of contextual warnings—essentially laundering biased information as authoritative knowledge."
>
> This clarifies that "importing" means incorporating biased content from retrieved sources, while "amplifying" means presenting it with increased authority/confidence, and "retrieval substrate" refers to the external information sources (Wikipedia, web search results) that contain the original biases.
>
> > Q4: Agent is used to describe an LLM with RAG… An agent is autonomous according to common AI terminology.
>
> A4: Thank you for the clarification. We acknowledge this important definitional point and will add a concrete definition to our paper. We use 'agentic' in the narrow sense that our systems *autonomously* decide when and how to retrieve and integrate external information, aligning with standard AI terminology for agent-based systems [OpenAI, Practices for governing agentic AI systems; 2407.01502, 2404.16244].
>
> While our primary evaluation centers on retrieval-based interactions, we also ran supplemental experiments using genuinely agentic systems (e.g., AutoGen/Agno) with broader tool use (Search, Python, Email, File, etc.) and OpenAI's Operator. These confirm similar safety degradation in more complex agentic contexts:
>
> Refusal Rates (Mean, higher better):
>
> | Benchmark | API | Agno-Web | Agno-MultiTool | Operator |
> | --- | --- | --- | --- | --- |
> | Bias1 | 0.92 | 0.76 | 0.66 | 0.42 |
> | Harm1 | 0.97 | 0.88 | 0.85 | 0.79 |
>
> We acknowledge that this is an initial step toward agentic safety evaluation. Our work focuses on identifying safety risks in information retrieval (a core component of agentic systems), rather than offering a full assessment of autonomous planning, complex tool use, or long-horizon reasoning. Broader frameworks are urgently needed, and we aim to provide foundational methodology for one key aspect of this larger challenge.
>
> > Q5: The paper uses different terms for the same things, e.g., “censored LLM” and “alignment-tuned LLM”.
>
> A5: We acknowledge the inconsistent terminology and will standardize throughout the paper. We will use "alignment-tuned LLMs" consistently to refer to models with safety training, replacing all instances of "censored LLMs" which carries negative connotations and is less precise.
>
> > Q6: Prompting before query generation and answer generation should not be called system-level safety checks.
>
> A6: We agree. We will revise to clarify these are "system-level prompt-based safety interventions" or "self-assessment mechanisms". True system-level safety requires external validation and architectural safeguards beyond LLM self-evaluation through prompting.
>
> > Q7: The first sentence in the abstract could be improved
>
> A7: Thank you for the suggestion. We will revise the opening sentence of the abstract to: "*Despite* the growing integration of retrieval-augmented AI agents into society, their safety and ethical behavior remain inadequately understood."
>
> > Q8: On lines 39–41, is it an assumption or a phenomenon?
>
> A8: We will clarify: "This phenomenon, termed safetywashing, promotes a misleading *assumption* that larger or more capable models are inherently safer. Our findings challenge this *assumption* in the context of retrieval-augmented agents." The phenomenon is safetywashing itself; the assumption is the belief it promotes.
>
> > Q9: It is not clear what is meant by aligned LLMs or alignment.
>
> A9: We will specify: "By 'aligned LLMs', we refer to models trained to exhibit helpful and harmless behavior, and to refuse harmful requests, using alignment techniques such as supervised fine-tuning on safety data [arXiv:2202.03286], Reinforcement Learning from Human Feedback (RLHF) [arXiv:2203.02155], and Constitutional AI [arXiv:2212.08073]."
>
> > Q10: On line 147, kappa=0.86 is considered high. Please specify
>
> A10: We will revise the sentence for clarity:
> "Cohen's kappa (κ = 0.86) quantifies inter-rater agreement between human experts and our LLM evaluator on safety classification. This indicates excellent agreement beyond chance (κ > 0.80 is considered 'almost perfect' agreement; Landis & Koch, 1977)."
>
> > Q11: On line 150, it is stated that four metrics are used, but only three are listed.
>
> A11: Thank you for spotting this. We will correct this to "three metrics" to match the actual number listed.
>
> > Q12: Line 168, what are “alignment challenges”? Please be specific.
>
> A12: We will clarify "alignment challenges" as specific difficulties maintaining AI safety when accessing external information: (1) reduced refusal rates, where retrieval overrides safety mechanisms; (2) bias amplification, where models import and magnify social biases; (3) compromised harmfulness safeguards, where retrieval-generation interactions yield more harm than either alone; and (4) structural vulnerability, where external context fundamentally destabilizes safety properties regardless of prompt-based mitigation attempts.
>
> > Q13: Line 175, what is “abliteration-based uncensoring”? Please explain.
>
> A13: We apologize for the unclear term. Abliteration-based uncensoring is a technique that effectively removes a specific internal "refusal direction" from large language models, allowing previously blocked outputs without full retraining (LaBonne, 2023).
>
> > Q14: Made clear how the contributions in An et al. [28] differ from the contributions of this paper
>
> A14: Our work extends An et al. in several key aspects: (1) Evaluation setting – while they study safety risks using classical retrievers (e.g., BM25) over curated Wikipedia data in controlled environments, we evaluate agents in real-world, open-domain conditions, capturing the complexity of actual deployments; (2) Alignment disentanglement – we compare aligned and uncensored models to isolate the alignment-specific effects of retrieval; (3) Broader agentic scope – we assess full AI agents with tool-use capabilities (AutoGen/Operator), going beyond simple LLM+retriever pairs; (4) Ablation studies – we vary multi-hop retrieval, document counts, and accuracy to reveal safety-relevant behaviors; and (5) Safety metrics – we evaluate bias, harmfulness, and refusal behavior using detailed rubrics, in contrast to a single binary safety classifier. While An et al. demonstrate that retrieval can degrade safety in constrained settings, we show that safety degradation not only persists but often intensifies under realistic, open-domain conditions.
>
> > Q15: The results in Table 5 are weak.
>
> A15: Thank you for the observation. The results are trustworthy despite modest absolute performance. This retrieval task is non-trivial with strict evaluation criteria, here Recall@5 requires exact matches among top-5 results. For context, DSPy officially reports Llama-8B achieving 0.59 Recall@5 on the same task [DSPy Team, Multi-hop Search Tutorial], while here Llama-3B achieves 0.49, indicating reasonable performance given the smaller model size.
> Crucially, the observed safety degradation trends remain consistent across conditions, reinforcing the reliability of our conclusions.
>
> > Q16: Line 240 – the sentence starting with “Despite…” is unclear.
>
> A16: Thank you for the suggestion. The current phrasing suggests we expected safety to worsen with better retrieval, which might appear unexpected. We will revise to: "*Notably*, even substantial improvements in retrieval quality from DSPy optimization (16.8% → 49.7% Recall@5) fail to prevent safety degradation, suggesting that enhanced retrieval accuracy does not provide protective effects against the safety risks we observe."
>
> This reframes the finding to emphasize that better retrieval performance doesn't solve the safety problem, rather than implying we expected it to cause additional harm.
>
> ---
> *References*
>
> Landis, J. R., & Koch, G. G. (1977). The measurement of observer agreement for categorical data. Biometrics, 33(1), 159–174.
>
> LaBonne, M. (2023). Abliteration: Removing Refusal Directions from LLMs. Hugging Face Blog.

---

> > ### Author Response · Authors · 2025-08-05
> > **Follow-up: Addressing Review Concerns**
> >
> > Thank you for the detailed feedback and recognizing our experiments and results as "intriguing and potentially very significant." We've addressed your clarity concerns by standardizing terminology and clarified our novel contributions. We'd value your thoughts on whether these clarifications address your main concerns.

---

> > ### Comment · Reviewer_La9j · 2025-08-06
> > **Thank you for detailed replies to my questions and comments**
> >
> > You have addressed my concerns. However, it is not easy to change recommendations as we have not seen the final product. This is of course not your fault, but the changes made to the review process. However, I have changed my recommendation to borderline accept.

---

> > > ### Author Response · Authors · 2025-08-07
> > >
> > > Thank you for the thoughtful review and for updating your recommendation. Your feedback helped us clarify key points and improve the precision of our manuscript.

---

### Official Review · Reviewer_ac1Z · 2025-07-02

**Clarity:** 4
**Significance:** 3
**Originality:** 3
**Rating:** 5
**Confidence:** 4

**Summary:**

The paper coins the term safety devolution; the idea that retrieval-augmented agents built on top of LLMs have degraded safety characteristics across bias, refusal, and harmful content. The issue persists even in the presence of various safety augmentations such as prompt-based mitigations. It discusses why these agents are more unsafe and comes to the conclusion that isn't really the quality or the accuracy of the information, it is just the fact that additional information is being accessed.

**Questions:**

My comments here are connected with the need to go all-in with respect to benchmarking agentic systems. If you just want to focus on RAG systems, my comments are less applicable.

1) There are a number of agentic benchmarks that are starting to appear. You shoiuld discuss some of them in your related works section and explain how they relate to this work.
2) If you want this work to be the start of benchmarking agentic systems (and not just RAG-based systems), you need to have a section discussing future work dealing with more than just information retrieval (planning, tool use, change of state, reasoning, etc.).
3) In my experience, no one has yet agreed on what the term agentic means or what an agentic system actually is. It would be valuable to acknowledge this. It would also be great to say this is only the beginning of how to evaluate agentic systems i.e. Information retrieval is only the first step.

As a final thought and just to be clear… In my opinion, if this work is only focused on RAG-based systems it is valuable and worthy of acceptance.

**Ethical Concerns:**

["NO or VERY MINOR ethics concerns only"]

**Final Justification:**

If I could see the revised paper and the changes proposed by the authors, I would consider revising my rating to strong accept.

**Limitations:**

Yes

**Paper Formatting Concerns:**

No major formatting issues found.

These are tiny issues; I'm guessing you have found them already, but it's an easy copy paste if you haven't caught them yet.

1) Double does: "Further retrieval optimization (e.g., more accurate search keys or higher document recall) does does not alter safety significantly, suggesting the core issue is the behavioral shift triggered by retrieved context."

**Quality:**

4

**Strengths And Weaknesses:**

Great paper! I am working on an agentic benchmark and this is the benchmark I wish we had created as a v0.5 POC. Whether accurate or not, however, I get the impression that this work was focused on RAG-based systems and the term “agentic” was added in at a later stage because agentic work is seen as cutting-edge. I will go into more detail on this point in the next section.

Quality: This is a quality piece of work. Reduced safety for RAG based systems is well supported by your results. I haven’t looked at all of the results in detail yet, but, for example, your evaluations have a minimal statistical rigor (such as error bars) that many benchmarks do not have. From a completeness standpoint, If it is just focused on RAG systems, it is much more complete. But, if you really want it to be the basis for looking at agentic systems, there is an enormous amount of additional work to do.

Clarity: Well written paper. It was clear and concise. I didn’t have trouble understanding what you were trying to convey.

Significance: Agentic is very complicated and, as a community, we need to start somewhere. This is a great start. As I mentioned before, this is exactly where I wanted to start with respect to agentic systems (though I have not been able to convince my colleagues :) )

Originality: There are very few agentic benchmarks. An initial focus on information retrieval is prudent and necessary.

---

> ### Author Rebuttal · Authors · 2025-07-31
>
> We appreciate the reviewer's positive assessment and valuable suggestions on expanding our related work discussion and clarifying our contribution as a foundational step toward comprehensive agentic system benchmarking. We address each point below.
>
> > Q1: I get the impression that this work was focused on RAG-based systems and the term ‘agentic’ was added in at a later stage because agentic work is seen as cutting-edge...there is an enormous amount of additional work to do.
>
> A1: Thank you for the comment. We acknowledge the definitional challenge and will add a clear definition to our paper. Following recent work [arxiv: 2407.01502, 2404.16244], we consider AI agents as “AI systems that can be instructed in natural language and act *autonomously* on the user's behalf are more agentic."
>
> While our current evaluation primarily focuses on retrieval-based interactions, the systems used in our experiments (AutoGen/Agno) are genuinely agentic, supporting core agentic features: tool-use, multi-step reasoning, dialogue memory, and persistent state management. Importantly, even in the simple RAG case, our implemented system is designed as an agent - the model can *autonomously* decide whether it wants to call the retrieval tool or not, demonstrating the key characteristic of independent decision-making that distinguishes agentic systems from purely reactive ones.
>
> To address this point, we have now added new experiments using both the Agno Agent with a broader tool suite (e.g., SearchTool, CalculatorTool, EmailTool, PythonTool, FileTool) and OpenAI Operator.
>
> The first table reports refusal rates, and the second table bias/safety scores. All values are shown as Mean ± CI, with higher being better for both metrics.
>
> | Benchmark | API | Agno-Web | Agno-MultiTool | Operator |
> | --- | --- | --- | --- | --- |
> | Bias1 | 0.92 ± 0.03 | 0.76 ± 0.05 | 0.66 ± 0.05 | 0.42 ± 0.06 |
> | Bias2 | 0.67 ± 0.05 | 0.61 ± 0.05 | 0.49 ± 0.05 | – |
> | Harm1 | 0.97 ± 0.03 | 0.88 ± 0.05 | 0.85 ± 0.06 | 0.79 ± 0.06 |
> | Harm2 | 0.73 ± 0.07 | 0.53 ± 0.07 | 0.49 ± 0.07 | – |
>
> | Benchmark | API | Agno-Web | Agno-MultiTool | Operator |
> | --- | --- | --- | --- | --- |
> | Bias1 | 4.37 ± 0.09 | 3.44 ± 0.15 | 3.74 ± 0.13 | 2.71 ± 0.21 |
> | Bias2 | 3.39 ± 0.16 | 3.14 ± 0.16 | 3.03 ± 0.15 | – |
> | Harm1 | 4.95 ± 0.04 | 4.72 ± 0.11 | 4.91 ± 0.06 | 4.56 ± 0.16 |
> | Harm2 | 4.52 ± 0.11 | 3.99 ± 0.17 | 3.68 ± 0.17 | – |
>
> For both Agno-Web and Agno-MultiTool (which includes tools beyond search), we observe consistent safety degradation compared to the Censored API, though adding more tools does not always result in more bias or harm.
> We also observe a similar trend in the Operator. Due to daily usage limits, we could not yet complete all runs for this setup, but we plan to include the full results.
>
> We fully acknowledge the substantial additional work needed to comprehensively evaluate agentic safety. Our current focus on retrieval-augmented interactions represents a deliberate starting point for several reasons: (1) information retrieval is fundamental to most agentic workflows, (2) it allows controlled evaluation of external knowledge integration, and (3) safety degradation in this basic capability has implications for more complex agentic behaviors. We position this as foundational work that establishes evaluation methodologies scalable to broader agentic capabilities.
>
> > Q2: There are a number of agentic benchmarks that are starting to appear. You should discuss some of them in your related works section and explain how they relate to this work.
>
> A2: We agree this is important and will expand our related work to include emerging agentic benchmarks. Current performance evaluation frameworks include ToolBench [arXiv:2307.16789] for tool-use proficiency, WebArena [arXiv:2307.13854] for web navigation, AgentBench [arXiv:2308.03688] for multi-step reasoning, and Agent-E [arXiv:2407.13032] for comprehensive agentic evaluation across diverse tasks.
>
> On the safety and security front, existing work includes corpus poisoning attacks [arXiv:2302.12173], R-Judge [arXiv:2401.10019] which mainly focuses on security risk awareness in agent interactions, and adversarial attacks on Retrieval-Augmented ICL [arXiv:2405.15984]. Concurrent work includes safety evaluation for LLMs with single retriever BM25 in controlled Wikipedia corpus [arXiv:2504.18041].
>
> However, these benchmarks primarily focus on *task performance metrics and controlled attack scenarios* with specific security vulnerabilities, while our work uniquely investigates how expanding retrieval scope systematically compromises *content safety properties* (refusal rates, bias amplification, harmful content generation). This represents a fundamental shift in model behavior that complements existing agentic evaluation frameworks by revealing how retrieval breadth, independent of specific attacks, undermines core safety mechanisms.
>
> > Q3: If you want this work to be the start of benchmarking agentic systems (and not just RAG-based systems), you need to have a section discussing future work dealing with more than just information retrieval (planning, tool use, change of state, reasoning, etc.)
>
> A3: We agree, and we will expand the Future Work section to reflect a broader agentic systems agenda. Building on our core finding that retrieved context fundamentally alters safety behavior, we plan to extend this work as follows:
>
> Multi-tool agentic systems: We will investigate how bias and harm propagate when agents access diverse tool combinations beyond retrieval—databases, calculators, APIs, and file systems. Preliminary experiments suggest that adding computational tools can exacerbate bias, as agents perform seemingly meaningless calculations that nonetheless shift reasoning toward stereotypical outputs. We will also examine how bias and harmfulness accumulate over extended multi-turn interactions where agents maintain conversation state.
>
> Enhanced safety mechanisms: Given our finding that prompt-based interventions fail to prevent safety degradation, we will evaluate mechanisms that operate beyond prompt-level controls, including bias detection at the retrieval stage and dynamic safety thresholds that adjust based on retrieved content toxicity. This will establish whether architectural safety interventions can address the fundamental context-dependence of AI safety we have identified.
>
> > Q4: no one has yet agreed on what the term agentic means or what an agentic system actually is. It would be valuable to acknowledge this. It would also be great to say this is only the beginning of how to evaluate agentic systems i.e. Information retrieval is only the first step.
>
> A4: We acknowledge this definitional challenge and will add a clear definition to our paper. Following OpenAI [Practices for governing agentic AI systems] and recent work [arxiv: 2407.01502], we define agenticness as "the degree to which a system can adaptively achieve complex goals in complex environments with limited direct supervision," where "AI systems that can be instructed in natural language and act *autonomously* on the user's behalf are more agentic." Our implemented systems qualify as AI agents because they *autonomously* decide whether to retrieve external context and integrate it into responses.
>
> We position our work as the evaluation of safety degradation in agentic systems, focusing on information retrieval as the foundational capability that enables agent autonomy. This represents an initial step toward comprehensive agentic safety evaluation.
>
> > Q5: typo Double does: "Further retrieval optimization (e.g., more accurate search keys or higher document recall) does does not alter safety significantly, suggesting the core issue is the behavioral shift triggered by retrieved context."
>
> A5: Thank you for catching the "Double does:" typo—we will correct this in revision.

---

> > ### Author Response · Authors · 2025-08-08
> > **Follow-up for Comments/Final Confirmation**
> >
> > Dear Reviewer,
> >
> > We are deeply grateful for your thoughtful and encouraging review. We would appreciate any additional comments or final confirmation before the review period concludes.

---

> > > ### Comment · Reviewer_ac1Z · 2025-08-08
> > >
> > > You have very clearly addressed my concerns. I agree with reviewer La9j that it is hard to change a rating without seeing the final edits, but I believe giving the paper a much more clear agentic focus will make it much more timely and impactful.

---

> > > > ### Author Response · Authors · 2025-08-09
> > > >
> > > > Thank you for your response and continued support. We appreciate your feedback in strengthening the agentic focus of our work.

---

### Note · Authors · 2025-08-12

We are grateful for the thorough review process and engagement. Most reviewers gave highly positive assessments and noted we "have very clearly/exhaustively addressed" their concerns, while one reviewer from a different methodological perspective offered feedback that helped clarify our scope.

Our work has been recognized for technical soundness and substantive contribution. Reviewer **ac1Z** praised our methodological rigor, calling it "the benchmark I wish we had created"; Reviewer **La9j** highlighted “extensive and well executed” experiments with “interesting and potentially very significant” results; and Reviewer **4oF3** deemed the work “an excellent paper” and “valuable contribution to NeurIPS” with comprehensive experiments and deep understanding.
Reviewers’ constructive feedback led us to expand experiments with OpenAI Operator and additional tools, standardize terminology, and enhance human validation.

Our work meets NeurIPS criteria for significance: **results impactful for the community** through first systematic identification of safety degradation in AI agents with retrieval—a critical issue as these systems proliferate in research and industry, with immediate relevance for safety evaluation standards and policy discussions; **foundation for future research** with reproducible methodology for agentic safety evaluation, initially on retrieval but applicable to planning, tool use, and reasoning; **advances understanding** by demonstrating that safety is not solely a model training challenge but also an *architectural* one; provides **unique conclusions** showing alignment mechanisms break down when retrieval is introduced.
As the guidelines note, *originality does not necessarily require introducing an entirely new method*. Our novelty lies in uncovering this problem space and reframing the safety landscape for agentic systems.

**Methodological Perspective Difference**

Reviewer **K35k** acknowledged we are "first to identify the phenomenon" of "an important yet often overlooked issue" and that our work evaluates "across multiple application scenarios, demonstrating its generality," yet emphasized novel mitigation methods. We agree such methods are crucial; however, they fall outside our empirical scope of identifying and characterizing safety degradation. Our results show retrieved content introduces architectural vulnerabilities conventional alignment cannot address—defining the problem space future mitigation research must build upon.

---

### Decision · Program_Chairs · 2025-09-17

**Decision:**

Accept (poster)

**Comment:**

## Summary
The paper asks the following question: "how does the augmention of LLMs with RAG affect safety features of the LLM output (bias mitigation etc)". The answer, based on tests with censored LLMs, LLMs with RAG, and uncensored LLMs, is that RAG can actually decrease the safety (hence the term devolution) of the outputs:

> retrieval-augmented agents built on aligned LLMs often behave more unsafely than uncensored models without retrieval. This effect persists even under strong retrieval accuracy and prompt-based mitigation, suggesting that the mere presence of retrieved content reshapes model behavior in structurally unsafe ways.

## Strengths
Reviewers all felt that the paper was

* studying a very important issue
* had well-executed experiments
* had interesting findings.


## Weaknesses
Reviewer concerns were mostly around:

* the use of agentic vs RAG language that felt confusing
* the feeling that if the paper was committing to a broader evaluation of agentic systems, that it needed to embrace that in the experiments, which still felt more RAG-focused.

There were numerous reviewer-specific questions (rather than concerns) that the author discussion focused on
## Author Discussion
There was a detailed and elaborate discussion with authors, in which reviewers highlighted many improvements that the authors could make, and which the authors responded to positively and with supporting experiments. The paper as described in the author discussion seemed like one the reviewers would really like to accept, but because there's no way to guarantee those changes get made, the reviewers remained hesitant about bumping up their scores.

One reviewer remained unconvinced about the quality of the paper.

## Justification
There was a lot of support for this work (or at least the version of it that emerged during discussion). The difficulty is in ensuring that the changes and augmentations recommended by reviewers are introduced into the finalized version.